# Tubulin Folding Cofactor TBCB is a Target of the *Salmonella* Effector Protein SseK1

**DOI:** 10.3390/ijms21093193

**Published:** 2020-04-30

**Authors:** Juan Luis Araujo-Garrido, Fernando Baisón-Olmo, Joaquín Bernal-Bayard, Francisco Romero, Francisco Ramos-Morales

**Affiliations:** 1Departamento de Genética, Facultad de Biología, Universidad de Sevilla, 41012 Sevilla, Spain; jaraujo@us.es (J.L.A.-G.); fernando.baisonolmo@gmail.com (F.B.-O.); jbbayard@us.es (J.B.-B.); 2Departamento de Bioquímica y Biología Molecular, Facultad de Ciencias Químicas y Farmacéuticas, Universidad de Chile, Santiago 1058, Chile; 3Departamento de Microbiología, Facultad de Biología, Universidad de Sevilla, 41012 Sevilla, Spain; frport@us.es

**Keywords:** TBCB, SseK1, tubulin, type III secretion, *Salmonella*, arginine N-glycosyltransferase

## Abstract

*Salmonella enterica* serovar Typhimurium is a human and animal pathogen that uses type III secretion system effectors to manipulate the host cell and fulfill infection. SseK1 is a *Salmonella* effector with glycosyltransferase activity. We carried out a yeast two-hybrid screen and have identified tubulin-binding cofactor B (TBCB) as a new binding partner for this effector. SseK1 catalyzed the addition of N-acetylglucosamine to arginine on TBCB, and its expression promoted the stabilization of the microtubule cytoskeleton of HEK293T cells. The conserved Asp-x-Asp (DxD) motif that is essential for the activity of SseK1 was required for the binding and modification of TBCB and for the effect on the cytoskeleton. Our study has identified a novel target for SseK1 and suggests that this effector may have a role in the manipulation of the host cell microtubule network to provide a safe niche for this pathogen.

## 1. Introduction

*Salmonella* is Gram-negative rod-shaped motile bacteria belonging to the family Enterobacteriaceae. *Salmonella* and many other Gram-negative symbionts and pathogens of animals and plants rely on type III secretion systems (T3SS) for their interactions with the host [1,2]. These systems usually work when bacteria are in contact with host cells and are able to translocate proteins, known as effectors, into the host cell cytoplasm [3]. *S. enterica* expresses two distinct T3SS for the manipulation of host cells, T3SS1 and T3SS2, that are encoded by genes located in the, *Salmonella* pathogenicity island (SPI) 1 and SPI2, respectively, and secrete more than 30 effectors. The T3SS1 provides a mechanism for host cell invasion that depends on the localized reorganization of actin filaments and the formation of membrane ruffles on the surface of host cells [4]. Once inside the *Salmonella* containing vacuole (SCV), the acidic pH and limitation of nutrients characteristic of this niche induce the expression of the T3SS2. This is a multifunctional virulence system that translocates effectors through the SCV to manipulate trafficking and maturation of this phagosome, providing a suitable niche for *Salmonella* intracellular survival and replication [5].

*S. enterica* serovar Typhimurium encodes three effectors, SseK1, SseK2 [6], and SseK3 [7], that belong to the NleB family of Asp-x-Asp (DxD)-containing glycosyltransferases [8]. This family of effectors includes NleB from *Citrobacter rodentium*, and NleB1, and NleB2 from enterohemorrhagic and enteropathogenic *Escherichia coli*. NleB1 adds N-acetylglucosamine (GlcNAc) to arginine in TNFR1-associated death domain protein (TRADD), FAS-associated death domain protein (FADD), and other death domain-containing proteins with an *N*-glycosidic linkage. This modification blocks the interaction between death domains, thereby disrupting inflammatory nuclear factor kappa B (NF-κB) signaling, caspase 8-dependent apoptosis, and necroptosis [9,10]. Similarly, during macrophage infection, both SseK1 and SseK3 regulate the activation of NF-κB negatively, and therefore, inhibit host cell death [11]. SseK1 promotes the GlcNAcylation of TRADD [9] and FADD [11], whereas SseK3 induces weaker arginine-GlcNAcylation of TRADD and no detectable modification of FADD, and there are conflicting reports about the biochemical activity of SseK2 [11,12]. SseK3 binds to the E3-ubiquitin ligase TRIM32 [13], but the significance of this interaction is unknown since this host protein is not modified by SseK3 and is not necessary for SseK3-dependent inhibition of NF-κB activation during the infection of macrophages [11]. In addition, a recent report using a mass spectrometry-based approach to enrich GlcNAcylated peptides from infected host cells showed that SseK3 modifies the death domains of receptors TNFR1 and TRAILR [14].

This work was undertaken to identify new host proteins targeted by SseK1. We show here that this effector binds tubulin-binding cofactor B (TBCB), transfers GlcNAc to this target, and modulates the stability of the microtubule cytoskeleton in a manner dependent on its glycosyltransferase activity.

## 2. Results

### 2.1. Salmonella SseK1 Interacts with Human TBCB in the Yeast Two-Hybrid System

The initial hypothesis of this work was that, after translocation through a T3SS, SseK1 would interact with host proteins that could also be substrates for the arginine glycosyltransferase activity of this *Salmonella* effector. In order to identify putative host interacting partners, SseK1 from the *S. enterica* serovar Typhimurium strain 14028 was fused to the DNA-binding domain of the bacterial LexA transcriptional factor using plasmid pLEX10 as a vector to yield plasmid pIZ2203. A yeast two-hybrid screen was carried out by transforming the *Saccharomyces cerevisiae* strain L40/pIZ2203 with a Jurkat (human lymphocyte cell line) cDNA library fused to the activation domain of the yeast transcriptional factor Gal4 using the vector pGAD1318. A total of 878 clones out of 7 × 10^5^ transformants were able to grow in synthetic medium lacking histidine, that was used to select for the interactions. The expression of a second reporter gene, *lacZ*, was also tested, and 180 clones demonstrated β-galactosidase activity (blue color in the presence of X-Gal). Plasmid derivatives of pGAD1318 were isolated from these positive clones, and a subset of them was reintroduced in yeasts L40/pIZ2203 and L40/pLEX10 and tested for reporter gene activation. Sequencing of the specific clones revealed that most of them carried cDNA from human TBCB. The interaction of SseK1 with TBCB in the yeast two-hybrid system is shown in Figure 1.

Additional clones were identified by PCR amplification with specific primers for TBCB and sequencing. Finally, a total of 130 clones corresponding to seven different genes demonstrated a specific interaction with SseK1 in the yeast two-hybrid system. A summary of the candidates identified is shown in Table 1.

### 2.2. SseK1 Copurifies and Coimmunoprecipitates with TBCB

Since most of the isolated clones corresponded to TBCB, we focused our study on the interaction of SseK1 with this host protein. To confirm the results obtained with the yeast two-hybrid approach, the bacterial glutathione S-transferase (GST) expression system was used to produce GST or GST-TBCB in a strain of *Salmonella* expressing a chromosomal SseK1-3xFLAG fusion (strain SV7071) [15]. Then, we performed affinity purification using glutathione-agarose beads to isolate GST proteins, and we further analyzed the presence of SseK1 by western blot. SseK1-3xFLAG copurified with GST-TBCB but not with GST (Figure 2A). Next, we performed coimmunoprecipitation assays to investigate if both proteins were able to interact in the more physiological context of the host cell. Human epithelial HeLa cells were cotransfected with pIZ2047, a derivative of plasmid pBABEpuro expressing SseK1-3xFLAG, and pIZ3423, a derivative of plasmid pCS2 expressing 3xHA-TBCB, or transfected with the later plasmid alone. SseK1-3xFLAG was immunoprecipitated from lysates of these cells and the copurification of 3xHA-TBCB was analyzed by immunoblot with anti-HA antibodies (Figure 2B). TBCB was coimmunoprecipitated with SseK1, but it was not precipitated in the absence of SseK1 (cells transfected with 3xHA-TBCB alone), showing the specificity of the interaction.

### 2.3. SseK1 Glycosylates TBCB in Vitro

Next, we wondered whether TBCB, being a new interacting partner of SseK1, could also be a new substrate for the glycosyltransferase activity of this *Salmonella* effector. To test this hypothesis, we incubated TBCB, purified as GST-fusion protein, with GST-SseK1 in the presence of UDP-GlcNAc, and detected substrate glycosylation by western blotting using a specific antibody against Arg-GlcNAc [16]. SseK1 glycosylated TBCB but not the N-terminal (amino acids 1–125) or C-terminal (amino acids 126–244) fragments of this protein, suggesting that the full-length protein is necessary to get a proper interaction and modification of this substrate (Figure 3). Self Arg-GlcNAcylation of SseK1 was also observed in these experiments, as previously reported for SseK3 [14]. SseK1 contains the conserved DxD catalytic motif found in glycosyltransferases. As expected, mutation of this motif to AAA caused the loss of both self-glycosylation and glycosylation of the host substrate TBCB (Figure 3).

### 2.4. SseK1 Glycosylates TBCB In Vivo

The results obtained in the previous section prompted us to examine the glycosylation of TBCB by SseK1 in the context of the mammalian host cell. Human HEK293T cells were transiently transfected with pIZ2336 (pEGFPC1-SseK1) or cotransfected with the same plasmid and pIZ3423 (pCS2-3xHA-TBCB), lysates were prepared in NP40 buffer 48 h after transfection and an immunoblot was carried out to detect Arg glycosylation. Several specific glycosylated proteins were detected in cells expressing GFP-SseK1 alone (Figure 4A). Interestingly an additional glycosylated band of the size expected for TBCB was detected in cells co-transfected with plasmids expressing GFP-SseK1 and 3xHA-TBCB. To ascertain that this band corresponded to glycosylated TBCB, lysates from transfected cells were further immunoprecipitated with an anti-HA antibody and analyzed by western blot with the anti-ArgGlcNAc antibody. A specific band was detected in lysates from cotransfected cells but not from cells expressing SseK1 only, confirming that TBCB was specifically glycosylated by SseK1 (Figure 4B).

### 2.5. SseK3 Interacts with TBCB

Since SseK2 and SseK3 share similar sequences and belong to the same family of effectors as SseK1, we decided to test the interaction of these two *Salmonella* effectors with human TBCB using the yeast two-hybrid system. The results were positive for SseK3 but not for SseK2 (Figure 5).

### 2.6. TBCB is not Glycosylated by SseK2 or SseK3

Next, we tested whether SseK2 or SseK3 were able to catalyze the glycosylation of TBCB. Western blot with anti-ArgGlcNAc antibody after the in vitro assay showed that TBCB was not glycosylated by SseK2 or SseK3, and only self-modification of SseK3 (but not SseK2) was detected (Figure 6). These results suggest that TBCB is specifically glycosylated by SseK1.

### 2.7. SseK1 Promotes Microtubule Stability

TBCB, together with TBCE, plays a role in microtubule biosynthesis [17], and overexpression of these cofactors leads to microtubule depolymerization [18]. The results presented above indicate that TBCB may interact with and be modified by SseK1, opening the possibility that this *Salmonella* effector may have an impact on microtubule dynamics. To test this hypothesis, we transfected HEK293T cells with plasmids expressing GFP-SseK1 or GFP alone, and we measured the acetylation level of α-tubulin by fluorescence microscopy since acetylation of α-tubulin correlates with microtubule stability [19,20,21]. The expression of SseK1 significantly increased the level of acetylated microtubules (Figure 7). Importantly, this effect was significantly reduced when cells were transfected with a plasmid expressing GFP-SseK1_AAA_, the catalytic mutant of this effector, suggesting that its GlcNAc transferase activity was involved in the alteration of the microtubule dynamics of the host cell.

## 3. Discussion

As mentioned in the Introduction, there are conflicting reports about the binding partners and preferred substrates of effectors belonging to the NleB/SseK family, suggesting that different methodologies are necessary to identify all possible host targets. Our approach to finding a new host substrate for the catalytic activity of SseK1 was based on a yeast two-hybrid screen, a useful technique used to identify protein interactions in vivo. Using that approach, we report here the identification of the tubulin-binding cofactor TBCB as a new binding partner for the *Salmonella* type III secretion effector SseK1. In addition, our in vitro and in vivo experiments demonstrated that TBCB is a substrate of the arginine-glycosylation activity of SseK1. 

Besides TBCB, our screen also identified other putative interacting partners of SseK1. However, these interactions need additional confirmation using independent techniques. Identification of binding partners for an effector, even if they are not substrates of its catalytic activity, as in the case of TRIM32 and SseK3, may also be important to understand as fully as possible the function of these virulence factors.

SseK1, SseK2, and SseK3 constitute a family of highly similar *Salmonella* effectors. SseK1 is 60% identical in amino acid sequence to SseK2 and SseK3, whereas SseK2 and SseK3 are 75% identical. The three proteins share a catalytic domain containing a DxD motif that is essential for the enzymatic function of most glycosyltransferases of the GT-A family [22]. Therefore, we tested if TBCB was also a substrate for SseK2 and SseK3. We detected a stable interaction between TBCB and SseK3. However, TBCB was not Arg-GlcNAcylated in the presence of this effector. This is in line with previous reports showing that there is some overlap, but also differential substrate specificity for these effectors [11,14] and that stable binding and catalytic activity are distinct aspects of the NleB/SseK family of effectors that are not always coupled. For instance, SseK1 is able to bind to and glycosylate glyceraldehyde 3-phosphate dehydrogenase (GAPDH) [12]; in contrast, SseK3 binds to TRIM32 [13] but is unable to promote its arginine GlcNAcylation [11].

Interestingly, SseK2 was unable to bind or modify TBCB. Although there is a report showing that SseK2 glycosylates FADD [12], other studies suggested that this effector lacks Arg-GlcNAc transferase activity [11,14]. Our results support the second possibility since we did not detect the apparent self-modification that was observed for SseK1 and SseK3. SseK2 might catalyze a different modification that is not detected by the anti-Arg-GlcNAc antibody, as suggested previously [11].

The best-characterized function of effectors of the NleB/SseK family is the inhibition of NF-κB signaling and of host cell death through the modification of death domain-containing proteins [9,10,11,14,23,24]. However, other ligands of NleB/SseK effectors have been reported [8]. For instance, *C. rodentium* NleB [25], enterohemorrhagic *E. coli* NleB1, and *S. enterica* SseK1 [12] are able to bind and glycosylate GAPDH. Glycosylation of GAPDH would disrupt its interaction with TRAF2 and TRAF3, leading to inhibition of NF-κB and type I IFN signaling [25,26]. A totally different role has been suggested for NleB1 from enteropathogenic *E. coli* through its interaction and modification of the α subunit of the hypoxia-inducible factor (HIF-1α) [27]. HIF-1 is a master regulator of cellular and systemic homeostatic response to hypoxia that activates transcription of genes involved in energy metabolism, angiogenesis, and apoptosis [28]. GlcNAcylation of HIF-1α enhances its transcriptional activity, leading to changes in host glucose metabolism. NleB1 also GlcNAcylated arginine on HIF-2α but did not enhance HIF-2α transcriptional activity [27].

NleB/SseK effectors also have a number of host binding partners whose GlcNAcylation has not been demonstrated. For instance, as mentioned above, TRIM32, the first binding partner described for SseK3 [13], is an E3 ubiquitin ligase that is not glycosylated by SseK3 [13] and is not required for the inhibition of NF-κB signaling during the infection of macrophages [11]. Human proteins DRG2, LRRC18, and POLR2E were found as partners of enterohemorrhagic *E. coli* NleB1 in a yeast two-hybrid screen [29], but these interactions were not confirmed by independent methods. A novel binding partner of this effector is MAP7 (ensconsin), which was identified by a quantitative proteomic approach and whose interaction was confirmed by coimmunoprecipitation [30].

In addition, NleB/SseK proteins may catalyze the glycosylation of endogenous bacterial proteins. A recent report has shown that NleB1 from enterohemorrhagic *E. coli,* NleB from *C. rodentium,* and SseK1 from *S. enterica*, can glycosylate the bacterial protein GshB (glutathione synthetase) [31]. This glycosylation enhances GshB-mediated production of glutathione, which provides resistance to oxidative stress, suggesting a new role for these effectors in promoting the survival of bacteria under these conditions.

The results shown here add a new binding partner for SseK1 and SseK3 and a new substrate for SseK1. These results also suggest the possibility of new effects on the host cell for SseK1. TBCB is one of five tubulin-binding cofactors termed TBCA, TBCB, TBCC, TBCD, and TBCE [32,33]. These cofactors interact with α- or β-tubulin to control tubulin dimer formation [34,35]. Both in vivo and in vitro experiments have shown that TBCE and TBCB form a complex together with α-tubulin leading to tubulin dissociation [18]. The upregulation of TBCB destabilizes the microtubules and causes low microtubule density [21,36]. The function of TBCB can be regulated by posttranslational modifications—gigaxonin [37], a putative substrate-specific adapter of an E3 ubiquitin ligase complex, promotes ubiquitination and proteasome-dependent degradation of TBCB and other microtubule-related proteins [37,38,39,40], PAK1 (p21-activated kinase 1) regulates microtubule dynamics by phosphorylating TBCB [41], and HILI (also known as PIWIL2), a member of the PIWI protein family, suppresses microtubule polymerization by inhibiting TBCB ubiquitination and degradation and reducing the phosphorylation level of TBCB induced by PAK1 [21]. Our results suggest that Arg-GlcNAcylation is a new posttranslational modification of TBCB catalyzed by SseK1. TBCB is a 244-aminoacid protein that contains an N-terminal UBL (ubiquitin-like) domain, a coiled-coil central domain and a C-terminal CAP-Gly (cytoskeleton-associated protein glycine-rich) domain [42]. Neither the N-terminal half of the protein, containing the UBL motif, nor the C-terminal half, containing the coiled-coil, and the CAP-Gly domains, were substrates for the activity of SseK1, suggesting that the full-length TBCB protein was necessary for the proper interaction with SseK1.

Since SseK1 catalyzes a new posttranslational modification of TBCB, we decided to study the impact of this effector on the microtubule cytoskeleton of host cells. Acetylation of the residue lysine 40 of α-tubulin has been associated with stable microtubules and thus used as a marker for microtubule stability [43]. Our experiments show that the expression of SseK1 in HEK293T cells increases the acetylation level of tubulin and that the conserved DxD catalytic motif is important for this effect, suggesting that the activity of this *Salmonella* effector affects microtubule stability.

Subversion of the cytoskeleton is a common way for *Salmonella* and other bacterial pathogens to manipulate the host cell [44]. *Salmonella* T3SS1 effectors SopE/E2, SopB, SipC, SipA, and SptP act in concert to alter the actin cytoskeleton promoting lamellipodia protrusions that facilitate bacterial entry through a trigger mechanism [45]. The maturation of the intracellular niche of *Salmonella*, the SCV, is accompanied by its translocation to a juxtanuclear position near the microtubule-organizing center. T3SS2 effectors SifA, SseF, SseG, and PipB2 contribute to the maintenance of this perinuclear localization and the formation of elongated tubular extensions along with the microtubule network through a balance between the recruitment of motor proteins kinesin and dynein [5]. SseJ is another *Salmonella* effector involved in endosomal tubulation [46] whose expression in host cells disrupts microtubule dynamics [47]. This effect may be explained by its binding to RhoA since this GTPase can stimulate microtubule stabilization via diaphanous-related formins [48]. Our findings suggest that SseK1 may stabilize microtubules through their interaction and modification of TBCB. Further experiments are needed to explore the possibility that SseK proteins may contribute to the positioning of the SCV, the elongation of *Salmonella* induced filaments, and the establishment of a safe replicative niche.

This work, in addition to highlighting the usefulness of the yeast two-hybrid system as a high throughput approach to describe new actors of the *Salmonella*-host biochemical dialogue, has allowed describing a new and unexpected aspect of the biological role of SseK1, not related with the previously reported host inflammation inhibition.

## 4. Materials and Methods 

### 4.1. Bacterial Strains, Yeast Strains, and Plasmids

Microbial strains and plasmids used in this study are described in Table 2. *S.* Typhimurium strains derived from the wild-type strain ATCC 14028. Transductional crosses using phage P22 HT 105/1 *int201* [49] were used for *Salmonella* strain construction [50].

### 4.2. DNA Amplification With the Polymerase Chain Reaction and Sequencing

Amplification reactions were carried out in a T100 Thermal Cycler (Bio-Rad, Hercules, CA, USA) using KAPA HiFi DNA polymerase (Kapa Biosystems, Wilmington, MA, USA) or MyTaq Red DNA polymerase (Bioline, Memphis, TN, USA) according to the instructions of the suppliers. Oligonucleotides are described in Table 3. Constructs were sequenced with an automated DNA sequencer (Stab Vida, Oeiras, Portugal).

### 4.3. Bacterial Culture

The standard culture medium for *S. enterica* and *E. coli* was lysogeny broth (LB). Solid LB contained agar 1.5% final concentration. Antibiotics were used at the following concentrations: kanamycin (Km), 50 μg/mL, ampicillin (Ap), 100 μg/mL. 

### 4.4. Yeast Two-Hybrid Methods

A human Jurkat cDNA library constructed in fusion with the activation domain of Gal4 in pGAD1318 was screened. *S. cerevisiae* strain L40 was sequentially transformed with pIZ2203 (pLEX10-SseK1) and the library by the lithium acetate procedure [56]. Transformants were plated on yeast drop-out medium lacking leucine, tryptophan, and histidine. Plates were incubated at 30 °C for three to eight days and then colonies were patched on the same medium and replica-plated on Whatman 40 filters placed on drop-out medium lacking leucine and tryptophan to test the β-galactosidase activity [57]. Positive clones were rescued, tested for specificity using empty pLEX10, and sequenced with the primer Gal4AD (Table 3). 

### 4.5. Cell Culture, Lysis, and Transfection

HeLa (human epithelial; ECACC no. 93021013) and HEK293T (human embryonic kidney SV40 transformed; ECACC no. 12022001) cells were cultured in Dulbecco’s modified Eagle’s medium (DMEM) supplemented with 10% fetal calf serum. 2 mM L-glutamine, 100 U/ml penicillin and 100 μg/mL streptomycin were included in the culture media. All cells were maintained in a 5% CO_2_ humidified atmosphere at 37 °C. For cell lysis, 2 × 10^7^ to 10^8^ cells per ml were incubated at 4 °C in NP40 buffer (10 mM Tris-HCl pH 7.4, 150 mM NaCl, 10% glycerol, 1% NP40, 1% aprotinin, 1 mM phenylmethylsulfonyl fluoride (PMSF), 1 μg/mL pepstatin, and 1 μg/mL leupeptin) for 20 min. The extract was centrifuged at 20000 g for 20 min, and the supernatant was stored at −80 °C. For transient transfection assays, 2–5 × 10^6^ HeLa cells/assay were resuspended in 200 μL of 15 mM HEPES-buffered serum-containing medium, mixed with 50 μL 210 mM NaCl containing 5–10 μg plasmid DNA and electroporated using a BTX Electrocell Manipulator 600 set at 240 V, 720 Ω, 950 μF. Cells were processed 24 h after electroporation. HEK293T cells were lipotransfected using the Xfect reagent (Takara Bio, Kusatsu, Japan) according to the manufacturer’s instructions and processed 48 h after transfection.

### 4.6. GST Fusion Proteins, Electrophoresis, and Immunoblot

Expression of the GST fusion proteins was induced by the addition of 1 mM isopropyl-β-D-thiogalactoside to bacteria containing pGEX-4T-1, pGEX-4T-3, or their derivatives and the fusion proteins were isolated from bacterial lysates by affinity chromatography with glutathione-agarose beads (Sigma-Aldrich, St Louis, MO, USA). For lysis, bacteria were sonicated in NP40 buffer. For some binding experiments, purification of fusion proteins was carried out in *Salmonella* strains expressing 3xFLAG-tagged proteins. The precipitates were washed six times in NP40 buffer followed by SDS-PAGE (polyacrylamide gel electrophoresis). The gel was blotted onto a nitrocellulose membrane and probed with anti-Flag M2 monoclonal antibodies (1:10000; Sigma-Aldrich, St Louis, MO, USA). Goat anti-mouse IRDye 800CW-conjugated antibodies (LI-COR Biosciences, Lincoln, NE, USA) were used as secondary antibodies. Bands were detected using the Odyssey Fc infrared imaging system (LI-COR Biosciences, Lincoln, NE, USA).

### 4.7. Coimmunoprecipitation Experiments

Lysates from HeLa cells cotransfected with plasmids pIZ2047 and pIZ3423, expressing SseK1-3xFlag and 3xHA-TBCB, respectively, were incubated for two hours with monoclonal anti-Flag antibodies, overnight with protein A/G plus-agarose beads (Santa Cruz Biotechnology, Dallas, TX, USA), and then centrifuged. The beads were washed five times in NP40 buffer. Proteins were eluted and dissolved into Laemmli sample buffer (50 mM Tris-HCl pH 6.8, 10% glycerol, 2% SDS, 0.0005% bromophenol blue) containing 5% β-mercaptoethanol, incubated at 95 °C for 5 min and subjected to SDS-PAGE. Proteins were transferred to a nitrocellulose filter and probed with anti-FLAG (Sigma) and anti-HA-peroxidase (clone 3F10, Roche).

### 4.8. In Vitro and in Vivo Glycosylation Assays

For in vitro assays, purified GST-fusion proteins were incubated for 4 h at room temperature in 50 mM Tris, pH 7.4, 10 mM MnCl_2_, 1 mM UDP-GlcNAc, 1 mM DTT. The reactions were terminated by boiling in a volume of Laemmli sample buffer. For in vivo assays, human HEK293T cells were transiently transfected with plasmids pIZ3423 (pCS2-3xHA-TBCB) and/or pIZ2336 (pEGFPC1-SseK1), lysates were prepared in NP40 buffer 48 h after transfection, and aliquots of these lysates were boiled with sample buffer. To immunoprecipitate TBCB, NP40 lysates were incubated for 2 h with monoclonal anti-HA antibodies, overnight with protein A/G plus-agarose beads, and then centrifuged. The beads were washed five times in NP40 buffer and proteins were eluted and dissolved into Laemmli sample buffer. Arg glycosylation was detected by immunoblot using an anti-Arg-GlcNAc monoclonal antibody (Abcam, Cambridge, UK). Goat anti-rabbit IRDye 680RD-conjugated antibodies (LI-COR Biosciences, Lincoln, NE, USA) were used as secondary antibodies. Bands were detected using the Odyssey Fc infrared imaging system (LI-COR Biosciences, Lincoln, NE, USA).

### 4.9. Mutagenesis

To generate point mutations in the conserved DXD motifs in *sseK1* or *sseK3*, pIZ2339 or pIZ3517 were used as templates for PCR amplification using primer pairs SseK1DXD223AAAfw/SseK1DXD223AAArv or SseK3DXD226AAAfw/SseK3DXD226AAArv. Products were digested with 1 μL of *Dpn*I (10 U/μL) for 1 h at 37 °C and used to transform *E. coli* DH5α. Mutations were verified by sequencing (Stab Vida, Oeiras, Portugal).

### 4.10. Immunofluorescence Microscopy

Cells were grown on slides for 24 h before an experiment. Cells were rinsed twice with PBS and incubated in 4% paraformaldehyde for 15 min at room temperature to fix the cells while preserving the GFP fluorescence. After aspiration of the fixative, preparations were rinsed three times with PBS for 5 min each and incubated in the blocking buffer for 60 min. Immunofluorescence analysis was performed on a Leica epifluorescence microscope. Cells were immunostained with anti-acetyl-α-tubulin (Lys40) (D20G3) (Cell Signaling Technology) primary antibody and goat anti-rabbit IgG secondary antibody with Alexa Fluor Plus 594 (Thermo Fisher Scientific). Acetyl-α-tubulin quantification was performed on epifluorescence microscopy images of single cells taken at identical magnification. Images were analyzed with the ImageJ software [58]. Statistical analysis was carried out with the Prism 6.0 software (GraphPad Software, San Diego, CA, USA) using the unpaired, non-parametric Kolmogorov–Smirnov test to determine the statistical significance of the differences between conditions.

## Figures and Tables

**Figure 1 ijms-21-03193-f001:**
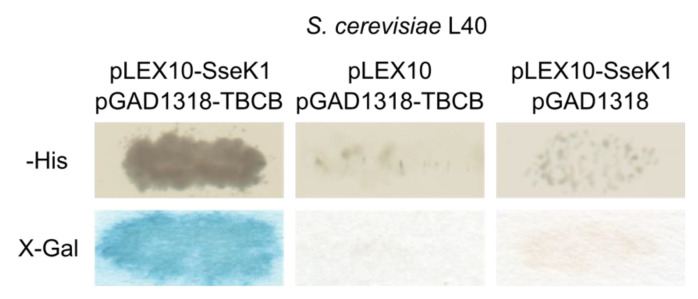
Human tubulin-binding cofactor B (TBCB) interacts with SseK1 in the yeast two-hybrid system. Strain L40 of *S. cerevisiae* was cotransformed with derivatives of plasmids pLEX10, to generate fusions with the DNA-binding domain of LexA, and pGAD1318, to generate fusions with the activation domain of Gal4, as indicated. The interaction between the two hybrid proteins is shown by the growth in the absence of histidine (-His), and the detection of blue color in the presence of X-Gal after a β-galactosidase filter assay. Empty vectors were used as negative controls.

**Figure 2 ijms-21-03193-f002:**
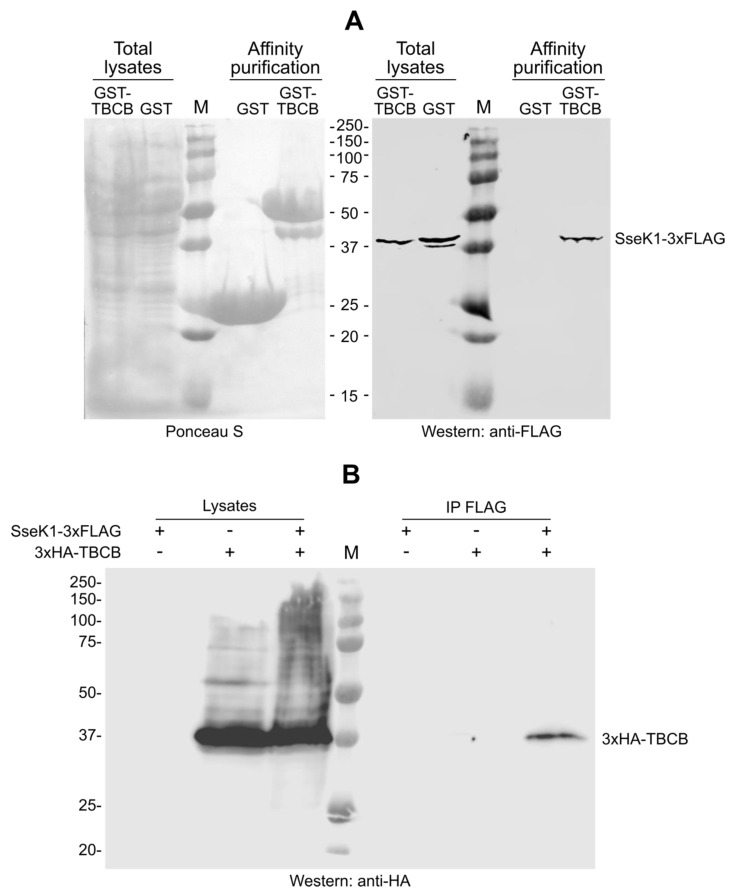
TBCB interacts with SseK1 in bacteria and host cells. (**A**) Bacterial lysates from strain SV7071 (14028 *sseK1*::3xFLAG) expressing glutathione S-transferase (GST) or GST-TBCB were prepared in NP40 buffer. Then, purified GST fusion proteins were blotted on nitrocellulose membranes, stained with Ponceau S red (left panel), and developed with monoclonal anti-FLAG (right panel). Aliquots of the original lysates were also included. The molecular weights of the marker bands (M) are indicated. (**B**) HeLa cells were transiently transfected with a derivative of pBABEpuro expressing SseK1-3xFLAG, a derivative of pCS2 expressing 3xHA-TBCB or cotransfected with the same plasmid and a derivative of pBABEpuro expressing SseK1-3xFLAG. NP40 lysates from 6 × 10^6^ transfected cells were subjected to immunoprecipitation (IP) with anti-FLAG monoclonal antibodies and, after washing, resolved by SDS-PAGE, transferred to nitrocellulose membranes, and developed with monoclonal anti-HA. Total lysates from 5 × 10^5^ transfected cells are included in the blot. M, molecular weight marker. Molecular weights in kDa are indicated on the left.

**Figure 3 ijms-21-03193-f003:**
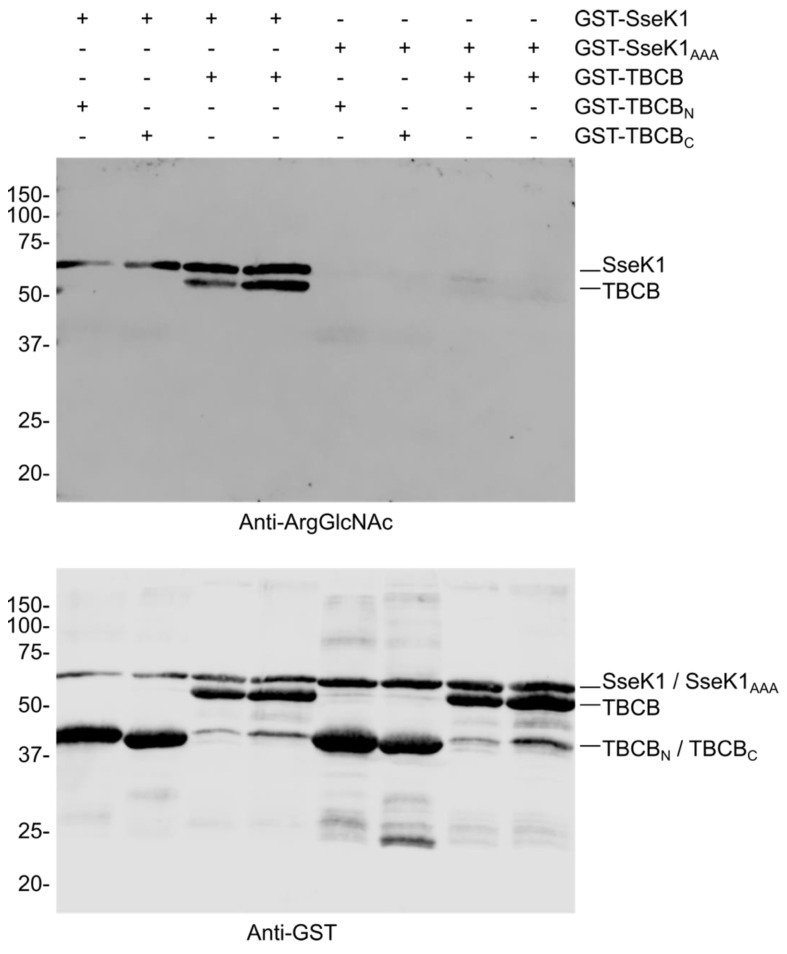
SseK1 glycosylates TBCB in vitro. Assay for SseK1 GlcNAc modification of TBCB using recombinant GST fusion proteins in the indicated combinations and 1 mM UDP-GlcNAc. SseK1_AAA_: SseK1(D223A/D225A). TBCB_N_: N-terminal fragment of TBCB (amino acids 1–125). TBCB_C_: C-terminal fragment of TBCB (amino acids 126–244). A representative immunoblot is shown incubated with anti-Arg-GlcNAc (upper panel). The same blot reincubated with anti-GST was used as the loading control (lower panel). Molecular weights in kDa are indicated on the left.

**Figure 4 ijms-21-03193-f004:**
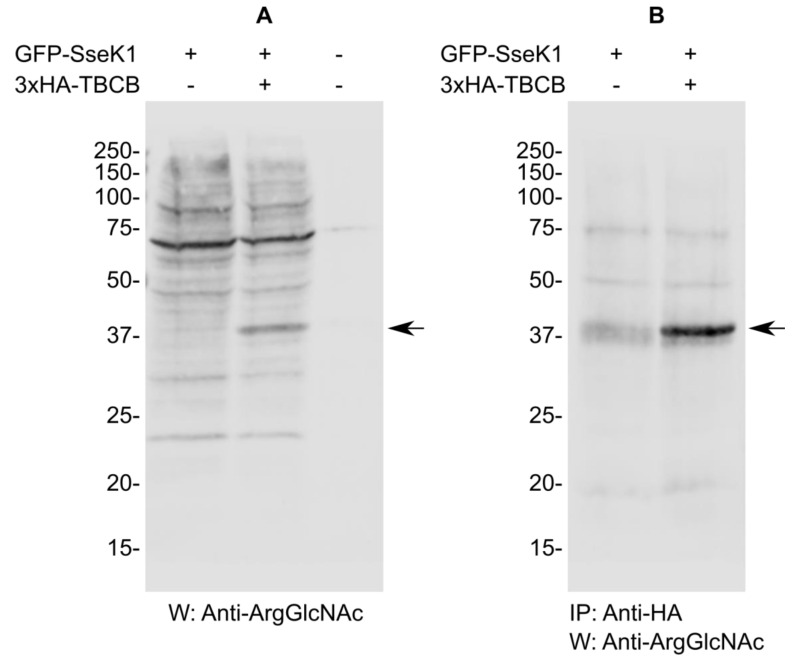
SseK1 glycosylates TBCB in vivo. (**A**) NP40 lysates from HEK293T cells transiently transfected with plasmids expressing GFP-SseK1 and/or 3xHA-TBCB, as indicated, were analyzed by immunoblot using anti-Arg-GlcNAc antibodies. (**B**) Lysates from HEK293T transiently transfected with a plasmid expressing GFP-SseK1 or with this plasmid and another plasmid expressing 3xHA-TBCB were subjected to immunoprecipitation (IP) with an anti-HA antibody. Precipitates were analyzed by western blot (W) with an anti-Arg-Glc-NAc antibody. Molecular weights in kDa are indicated on the left. The arrow indicates the band corresponding to TBCB.

**Figure 5 ijms-21-03193-f005:**
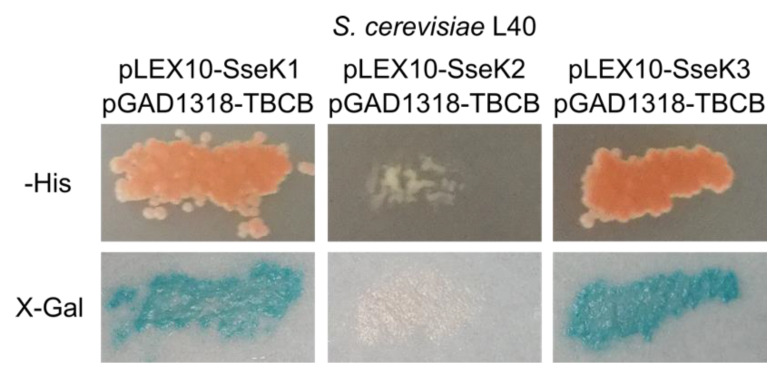
SseK3 interacts with TBCB in the yeast two-hybrid system. Strain L40 of *S. cerevisiae* was cotransformed with derivatives of plasmids pLEX10, to generate fusions of SseK1, SseK2, or SseK3 with the DNA-binding domain of LexA, and pGAD1318, to generate a fusion of TBCB with the activation domain of Gal4, as indicated. The interaction between the two hybrid proteins is shown by the growth in the absence of histidine (-His), and detection of blue color in the presence of X-Gal after a β-galactosidase filter assay.

**Figure 6 ijms-21-03193-f006:**
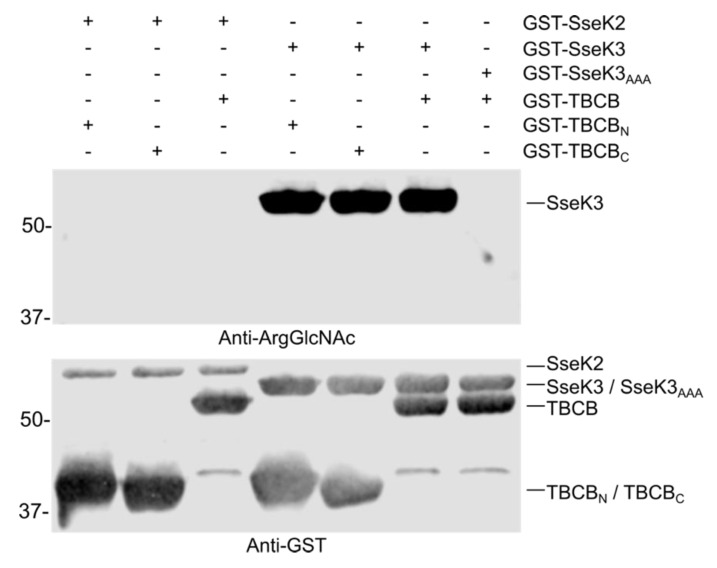
SseK2 and SseK3 are unable to glycosylate TBCB in vitro. An assay for GlcNAc modification of TBCB in the presence of SseK2 or SseK3 using recombinant GST fusion proteins in the indicated combinations and 1 mM UDP-GlcNAc. SseK3_AAA_: SseK3(D226A/D228A). TBCB_N_: N-terminal fragment of TBCB (amino acids 1–125). TBCB_C_: C-terminal fragment of TBCB (amino acids 126–244). An immunoblot incubated with anti-Arg-GlcNAc is shown (upper panel). The same blot was reincubated with anti-GST (lower panel). Molecular weights in kDa are indicated on the left.

**Figure 7 ijms-21-03193-f007:**
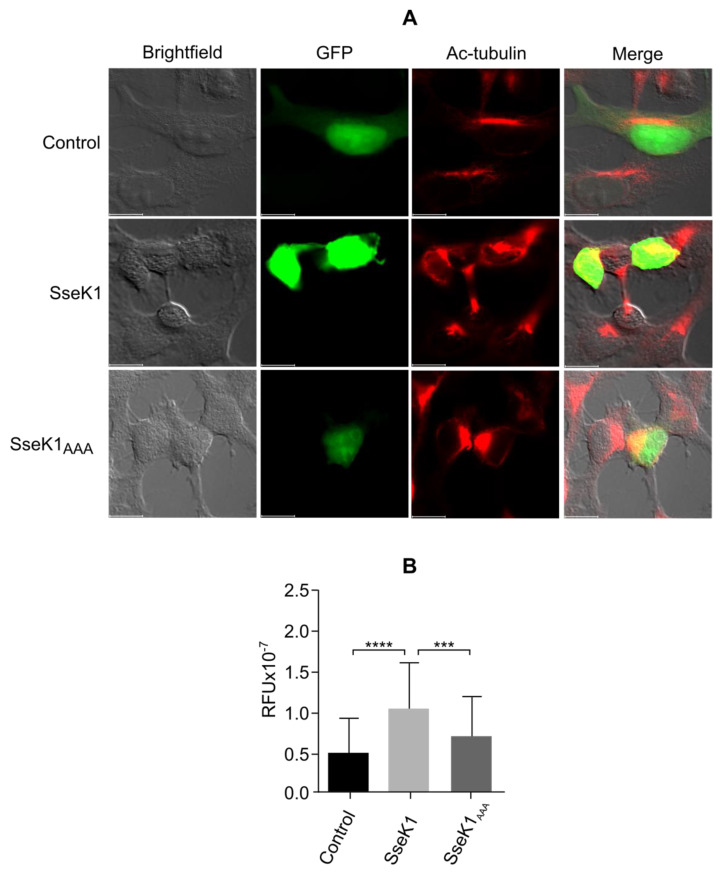
SseK1 increases the acetylation level of tubulin in HEK293T cells. (**A**) Representative fluorescence microscopy images showing GFP (Control) or GFP fusion proteins (GFP-SseK1 or GFP-SseK1_AAA_,) in green, and acetyl-α-tubulin in red. HEK293T cells were transfected with derivatives of plasmid pEGFPC1, and 48 h after transfection cells were fixed with 4% paraformaldehyde and stained with anti-acetyl-α-tubulin rabbit monoclonal antibody and Alexa Fluor Plus 594-conjugated anti-rabbit secondary antibody. (**B**) Quantification of acetyl-α-tubulin from immunofluorescence images of cells transfected with GFP (Control) or GFP fusion proteins (GFP-SseK1, GFP-SseK1_AAA_). The mean and standard deviation from 100 cells are represented for each condition. RFU, relative fluorescence units. ***, *P* < 0.001; ****, *P* < 0.0001, for the indicated comparisons using a Kolmogorov–Smirnov test.

**Table 1 ijms-21-03193-t001:** Candidate host partners of SseK1 identified in a yeast two-hybrid screen.

Gene	Number of Clones	Description of the Product	Amino Acids Included in the Identified Clones
*TBCB*	116	Tubulin-binding cofactor B	1–244 4–244 29–244
*ZBTB16*	7	Zing finger and BTB domain-containing protein 16	420–673 448–673 552–673
*H3F3B*	3	Histone H3.3	1–136 40–136
*H3F3A*	1	Histone H3.3	18–136
*CENPA*	1	Histone H3-like centromeric protein A	1–140
*HIRIP3*	1	HIRA-interacting protein 3	340–556
*XRCC6*	1	X-ray repair cross-complementing protein 6	335–609

**Table 2 ijms-21-03193-t002:** Bacterial strains and plasmids used in this study.

Strain/Plasmid	Relevant Characteristics	Source/Reference
***Escherichia coli***		
BL21(DE3)	F^−^ *ompT gal dcm lon hsdS_B_* (r^−^ m^−^; *E. coli* B strain), with DE3, a λ prophage carrying the T7 RNA *pol* gene	Stratagene
DH5α	*supE44 ∆lacU*169 (Ø80 *lacZ*∆M15) *hsdR17 recA1 endA1 gyrA96 thi-1 relA1*	[51]
HB101	*F^-^ mcrB mrr hsdS20 (r_B_^−^ m_B_^−^) recA13 leuB6 ara-14 proA2 lacY1 galK2 xyl-5 mtl-1 rpsL20(Sm^R^) glnV44 λ^−^*	[52]
***Salmonella enterica* serovar Typhimurium^a^**		
14028	Wild type	ATCC
SV7071	*sseK1*::3xFLAG, Km^r^	[15]
***Saccharomyces cerevisiae***		
L40	*MAT* *α trp1 leu2 his3 LYS2::lexA-HIS3 URA3::lexA-lacZ*	[53]
***Plasmids***		
pEGFPC1	GFP fusion vector, Km^r^	Clontech
pGEX-4T-1	GST fusion vector, Ap^r^	GE Healthcare
pGEX-4T-3	GST fusion vector, Ap^r^	GE Healthcare
pGAD1318	Yeast two-hybrid vector, Ap^r^	[54]
pLEX10	Yeast two-hybrid vector, Ap^r^	[55]
pIZ2047	pBABEpuro-SseK1-3xFLAG	This work
pIZ2203	pLEX10-SseK1	This work
pIZ2260	pGAD1318-TBCB	This work
pIZ2336	pEGFPC1-SseK1	This work
pIZ2339	pGEX-4T-3-SseK1	This work
pIZ3405	pLEX10-SseK2	This work
pIZ3406	pLEX10-SseK3	This work
pIZ3423	pCS2-3xHA-TBCB	This work
pIZ3509	pGEX-4T-1-TBCB	This work
pIZ3510	pGEX-4T-1-TBCB(1–125)	This work
pIZ3511	pGEX-4T-1-TBCB(126–244)	This work
pIZ3516	pGEX-4T-1-SseK2	This work
pIZ3517	pGEX-4T-1-SseK3	This work
pIZ3518	pGEX-4T-3-SseK1(D223A/D225A)	This work
pIZ3519	pGEX-4T-1-SseK3(D226A/D228A)	This work
pIZ3522	pEGFPC1-SseK1(D223A/D225A)	This work

**Table 3 ijms-21-03193-t003:** Oligonucleotides used in this study.

Oligonucleotide/Use	Sequence 5’-3’
**Construction of pIZ2047**	
SseK1pBABE_5’	GTCAGGATCCGCCGCCACCATGATCCCACCATTAAATAG
3FlagSal3’	CTGAGTCGACTTACTATTTATCGTCGTCATC
**Construction of pIZ2203**	
SseK1pLex10EcoRIfw	ATGCGAATTCATGATCCCACCATTAAATAG
SseK1pLex10SalIrev	ATGCGTCGACCTACTGCACATGCCTCGCCC
**Construction of pIZ2336**	
SseKIGFPCEcoRIfw	ATCGGAATTCGATGATCCCACCATTAAATAGATATG
SseKIGFPCBamHIrev	ATCGGGATCCATTTCCGCTACTGCACATGC
**Construction of pIZ3405**	
SseK2ecofw	GATCGAATTCATGGCACGTTTTAATGCCGC
SseK2salrv	ACGTGTCGACTTACCTCCAAGAACTGGCAG
**Construction of pIZ3406**	
SseK3ecofw	ATGCGAATTCATGTTTTCTCGAGTCAGAGG
SseK3salrv	ATGCGTCGACTTATCTCCAGGAGCTGATAG
**Construction of pIZ3423**	
TBCBecoRIfw	ATCGGAATTCGAGGTGACGGGGGTGTCGGC
TBCBxbaIrev	ATCGTCTAGATCATATCTCGTCCAACCCG
**Construction of pIZ3509**	
TBCBEcoRIfw	ATCGGAATTCGAGGTGACGGGGGTGTCGGC
TBCBSTOPxhoIrev	ATCGCTCGAGGTCATATCTCGTCCAACCCG
**Construction of pIZ3510**	
TBCBEcoRIfw	ATCGGAATTCGAGGTGACGGGGGTGTCGGC
TBCB125STOPxhoIrev	ATCGCTCGAGTCACAGGAAAGAGCGGACCGTGTC
**Construction of pIZ3511**	
TBCB126EcoRIfw	ATCGGAATTCAAGCGCAGCAAGCTCGGCCG
TBCBSTOPxhoIrev	ATCGCTCGAGGTCATATCTCGTCCAACCCG
**Construction of pIZ3521**	
Ssek3pEGFPC1ecoriFW	ATCGGAATTCTATGTTTTCTCGAGTCAGAGG
SseK3salrv	ATGCGTCGACTTATCTCCAGGAGCTGATAG
**Identification of candidates carrying TBCB**	
TBCBfw	GCTCCTACCCTGTAGATGACG
TBCBrev	CATATCTCGTCCAACCCGTAG
**Mutagenesis of SseK1**	
SseK1DXD223AAAfw	GGTGTATATATCTTGCTGCTGCTATGATTATCACGG
SseK1DXD223AAArev	CCGTGATAATCATAGCAGCAGCAAGATATATACACC
**Mutagenesis of SseK3**	
SseK3DXD226AAAfw	GGCTGCATATATCTTGCTGCTGCTATGTTACTTACAGG
SseK3DXD226AAArev	CCTGTAAGTAACATAGCAGCAGCAAGATATATGCAGCC
**Sequencing of two-hybrid screen candidates**	
Gal4AD	TACCACTACAATGGATG

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
