# Peer review of "Tubulin Folding Cofactor TBCB is a Target of the Salmonella Effector Protein SseK1"

_ijms, 2020, doi:10.3390/ijms21093193_

Round 1

Reviewer 1 Report

This is a clearly written paper that definitively shows that the Salmonella effector SseK1 binds and glycosylates the tubulin folding cofactor TBCB. The data are rigorous and presented clearly. Overall, the authors provide evidence for a potentially-important new substrate for this bacterial glycosyltransferase. I have very few suggestions for improvement and believe that the work has sufficient merit for publication in this journal.

Edits:

1 - line 41, change 'belongs' to 'belong'

2 - Figure 2B, SseK1-3xFLAG legend -- the bottom of this label is partially cut-off and the authors may need to adjust this line slightly in their illustration software

3 - section 2.8. greater impact would be provided if the authors were to formally assess microtubule dynamics as a function of SseK1 transfection, rather than just rely on tubulin acetylation as a correlate. However, I leave this as a mere suggestion for the authors to consider.

Author Response

We thank the reviewers for their constructive criticisms that help to improve the quality of the manuscript. In this new version we have introduced modifications to address all the suggestions provided by the reviewers and the editor. All the changes are highlighted in the manuscript in red. Specific changes are as follows:

Responses to reviewer 1:

This is a clearly written paper that definitively shows that the Salmonella effector SseK1 binds and glycosylates the tubulin folding cofactor TBCB. The data are rigorous and presented clearly. Overall, the authors provide evidence for a potentially-important new substrate for this bacterial glycosyltransferase. I have very few suggestions for improvement and believe that the work has sufficient merit for publication in this journal.

Edits:

1 - line 41, change 'belongs' to 'belong'

Response: -Typo corrected as suggested in line 41.

2 - Figure 2B, SseK1-3xFLAG legend -- the bottom of this label is partially cut-off and the authors may need to adjust this line slightly in their illustration software

Response: -We thank the reviewer for the observation. This problem does not exist in the original figure or in the word file. Maybe it was just the result of the pdf conversion. Please, look at the revised version of the word file.

3 - section 2.8. greater impact would be provided if the authors were to formally assess microtubule dynamics as a function of SseK1 transfection, rather than just rely on tubulin acetylation as a correlate. However, I leave this as a mere suggestion for the authors to consider.

Response: -We thank the reviewer for the suggestion. The necessary experiments, however, are out of the scope of this paper, but we will certainly take into account this suggestion in our future research work.

Reviewer 2 Report

This is a well-written paper on a very interesting topic. Authors have launched a screening based on a two-hybrid system assay to find new interaction partners of the SseK1 effector in the host cell. I have only a few comments:

-Line 75, Figure 1 is not including any of the information described on this sentence.

-The results shown in Figure 7 are not very convincing. The differences observed in between SseK1 and SseK1AAA are quite modest. In particular, this reviewer is not observing any significant differences in between all figures included on panel A in the column Ac-Tubulin. In addition, the GFP-SseK1 fusion proteins seem distributed through the whole cell, and there is no clear indication of interaction in between tubulin and SseK1 in these figures. Overall, these data seem quite preliminary. Therefore, the authors should substantiate these results with better figures showing clear co-localization events and with a more robust quantification of the acetylation levels of alpha-tubulin, perhaps by western blotting.

-Lines 212 and 279, please show the data or remove these sentences.

Round 2

Reviewer 2 Report

No further comments.